# Conditional repair by locally switching the thermal healing capability of dynamic covalent polymers with light

Anne Fuhrmann[1], Robert Göstl[1], Robert Wendt[1], Julia Kötteritzsch[2,3], Martin D. Hager[2,3], Ulrich S. Schubert[2,3], Kerstin Brademann-Jock[4], Andreas F. Thünemann[4], Ulrich Nöchel[5], Marc Behl[5] & Stefan Hecht[1]

Healable materials could play an important role in reducing the environmental footprint of our modern technological society through extending the life cycles of consumer products and constructions. However, as most healing processes are carried out by heat alone, the ability to heal damage generally kills the parent material's thermal and mechanical properties. Here we present a dynamic covalent polymer network whose thermal healing ability can be switched 'on' and 'off' on demand by light, thereby providing local control over repair while retaining the advantageous macroscopic properties of static polymer networks. We employ a photoswitchable furan-based crosslinker, which reacts with short and mobile maleimide-substituted poly(lauryl methacrylate) chains forming strong covalent bonds while simultaneously allowing the reversible, spatiotemporally resolved control over thermally induced de- and re-crosslinking. We reason that our system can be adapted to more complex materials and has the potential to impact applications in responsive coatings, photolithography and microfabrication.

[1] Department of Chemistry & IRIS Adlershof, Humboldt-Universität zu Berlin, Brook-Taylor Straße 2, 12489 Berlin, Germany. [2] Laboratory of Organic and Macromolecular Chemistry (IOMC), Friedrich Schiller University Jena, Humboldtstraße 10, 07743 Jena, Germany. [3] Jena Center for Soft Matter, Friedrich Schiller University Jena, Philosophenweg 7, 07743 Jena, Germany. [4] Federal Institute for Materials Research and Testing, Unter den Eichen 87, 12205 Berlin, Germany. [5] Institute of Biomaterial Science and Berlin-Brandenburg Center for Regenerative Therapies (BCRT), Department Active Polymers, Helmholtz-Zentrum Geesthacht, Kantstraße 55, 14513 Teltow, Germany. Correspondence and requests for materials should be addressed to S.H. (email: sh@chemie.hu-berlin.de).

Future technologies require smart materials with advanced properties including responsiveness to external stimuli and in particular the ability to autonomously repair inflicted damage[1–3]. Such ability to heal defects is typically achieved by the release of microencapsulated healing agents[4,5] or by the introduction of reversible connections in the polymer architecture, either in a non-covalent fashion, such as in supramolecular polymers[2,6], or by using dynamic covalent chemistry[7–9]. One particularly clever approach involves the implementation of reversible polymer networks[10,11] as they offer outstanding mechanical strength and thermal properties that are readily modified by the nature of the connecting dynamic bonds and the crosslinking density. In this context, the reversible Diels–Alder reaction of furans with maleimides is extremely popular[11,12] as it does not require additional catalyst or produce by-products leading to static covalent crosslinks at ambient conditions. At elevated temperatures these polymer networks, however, become dynamic and reorganization within the material to repair inflicted damage is enabled. In these thermoresponsive, reversibly cross-linked polymers damage repair is solely triggered and driven by heat acquiring the ability to heal at the expense of the desirable thermal and hence mechanical properties of a static polymer network. Consequently, conditional healing of the materials under load and in specific areas is difficult to achieve[13].

Other materials try to circumvent this issue by activation of their healing or repairing ability through chemical signals, such as pH (ref. 14) or redox changes[15], or physical stimuli, such as mechanical stress[16]. However, as it would be highly attractive to selectively address and reversibly switch the dynamic covalent bond 'on' or 'off', light arguably presents itself as the ideal stimulus: It offers the possibility to activate molecules with a high spatiotemporal as well as energetic resolution without the need to invasively alter the material properties thus rendering photo-switches the ideal molecules to locally control thermal healing behaviour. In stark contrast, solely photo-driven approaches require continuous light irradiation to power the healing process[17–20].

Our conditional healing approach relies on a set of photo-switchable reactants[21] whose ability to undergo a thermally reversible crosslinking reaction ($\mathbf{X_{ON}} + \mathbf{P} \rightleftharpoons \mathbf{X@P_{ON}}$)—that is, to connect and disconnect polymer chains ($\mathbf{P}$)—can be controlled by light to switch between a dynamic (ON) and a static system (OFF) (Fig. 1a). By illumination with ultraviolet-light, both the free crosslinker $\mathbf{X_{ON}}$ as well as the cross-linked polymer $\mathbf{X@P_{ON}}$ can be converted to their respective locked states ($\mathbf{X_{OFF}}$ and $\mathbf{X@P_{OFF}}$) from which the dynamic covalent reaction is inhibited. From there, visible light can selectively induce the re-activation of the polymer's dynamic character leading to a reversibly switchable dynamic covalent material. We designed our system based on photoswitchable furans since Branda and coworkers[22,23] as well as our own group[24,25] have recently shown control over reversible Diels–Alder reactions with maleimide. The

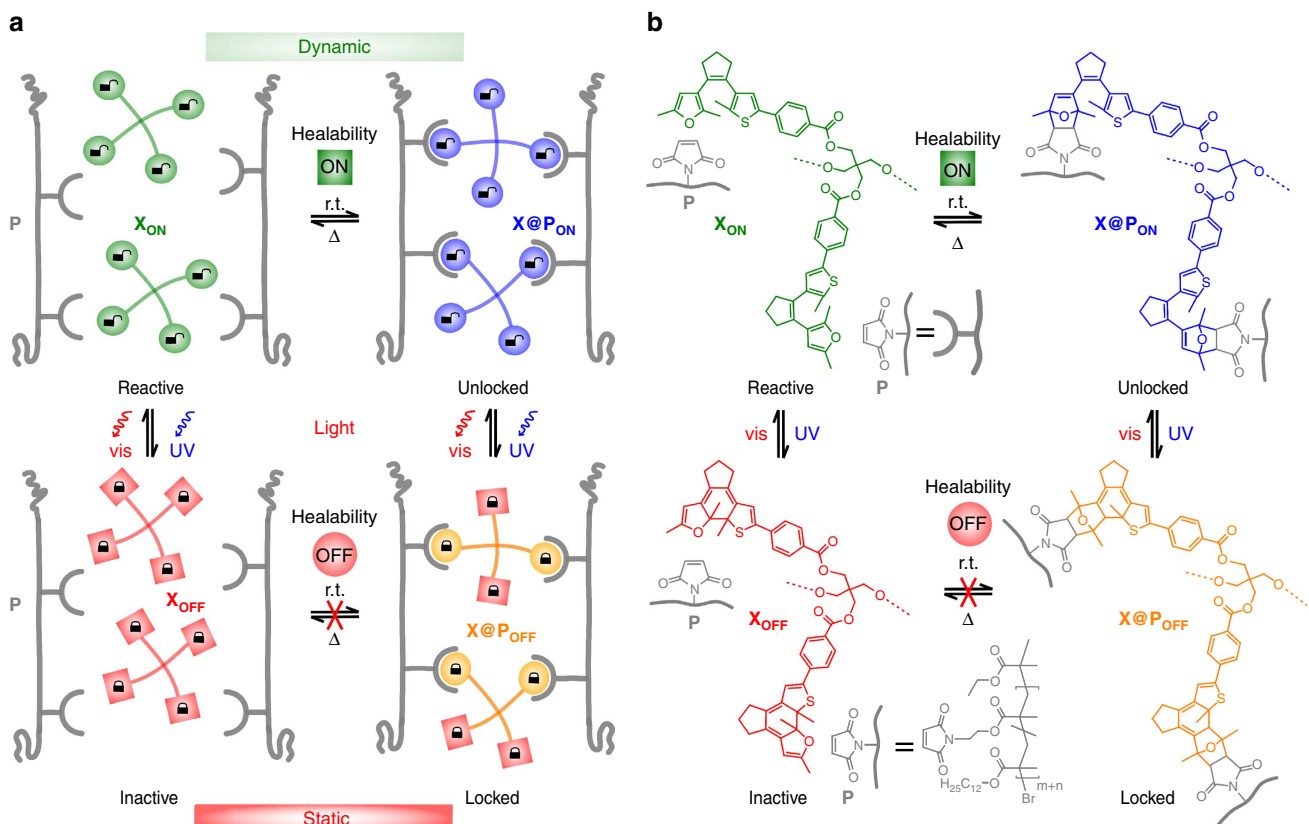

**Figure 1 | Switching the thermal healability of dynamic polymer networks by light. (a)** Schematic representation of a photoswitchable crosslinker $\mathbf{X_{ON}}$ in its reactive state and polymer $\mathbf{P}$ carrying complementary reactive groups forming the dynamic covalent polymer network $\mathbf{X@P_{ON}}$, which can be de-crosslinked on heating. On illumination with ultraviolet-light the crosslinker $\mathbf{X}$ as well as the polymer network $\mathbf{X@P}$ are reversibly transformed into their inactive state ($\mathbf{X_{OFF}}$ and $\mathbf{X@P_{OFF}}$, respectively), where both crosslinking as well as de-crosslinking are inhibited. **(b)** Chemical structures of the tetrafuryl-substituted diarylethene crosslinker, which in its activated form $\mathbf{X_{ON}}$ reacts with maleimide-functionalized poly(lauryl methacrylates) via the Diels–Alder reaction (for simplicity only two of four crosslinking units are shown). On heating the network is de-crosslinked due to the retro Diels–Alder reaction. Illumination with ultraviolet-light induces 6π-electrocyclization to the ring-closed isomers of the deactivated crosslinker $\mathbf{X_{OFF}}$ and the polymer $\mathbf{X@P_{OFF}}$, respectively. This process can be reversed and the crosslinker as well as the network can be activated by illumination with visible light.

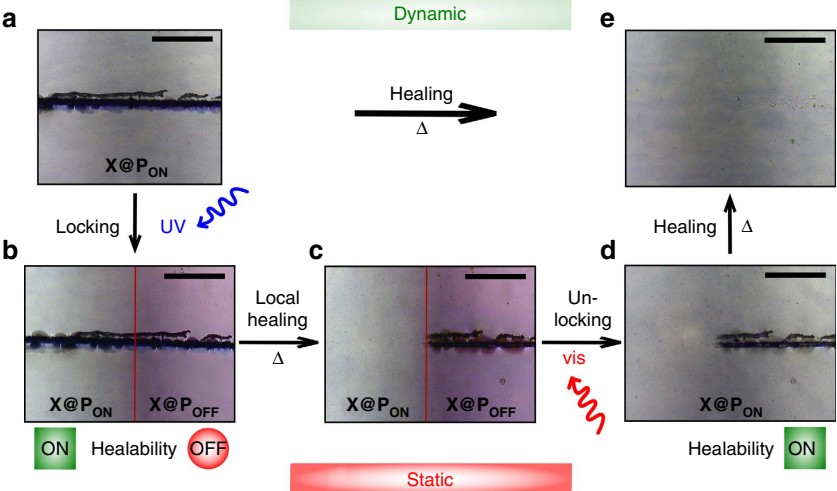

**Figure 2 | Visualization of macroscopic scratch healing with and without light control.** Micrographs of a scratched thin film of **X@P$_{ON}$**, which is healed on heating to 124 °C for 5 min in vacuum (**a,e**) (dynamic system). After illuminating the right side of the scratched film with a 365 nm LED for 30 min under an argon atmosphere (**a,b**), the scratch is fixated on heating to 124 °C for 5 min in vacuum (**X@P$_{OFF}$, b,c**) whereas the healing ability on the left side (**X@P$_{ON}$**) remains switched 'on'. The healing ability of **X@P$_{OFF}$** is regenerated after illumination with a 460 nm LED for 105 min under an argon atmosphere (**c,d**) and after heating to 124 °C for 5 min in vacuum (**d,e**). Scale bar, 0.3 mm.

furan-containing diarylethene photoswitches employed in these works are especially well suited for this purpose as they guarantee the bistability of the switchable material and can be addressed selectively and independently using ultraviolet and visible light[26].

## Results

**Design and synthesis of crosslinker and linear polymer.** For efficient crosslinking, more than one reactive furyl moiety is necessary and hence we utilize a tetrafunctional motif **X** inspired by Wudl and coworkers[11] minimizing the amount of crosslinker and hence optical density to warrant maximum light penetration. This is an intrinsic advantage over photocontrollable linearly connected polymer chains where the photochromic compounds form the polymer backbone[23]. Moreover, the connection of the diarylethene to the pentaerythritol core via electron-withdrawing ester groups guarantees high fatigue resistance and photo-chemical efficiency[27], while retaining the beneficial spectral separation of the different isomers (Fig. 1b, for synthesis see Supplementary Methods). A further benefit over already reported systems[23] is the ability to switch 'off' the unreacted crosslinker adding the possibility to activate the dynamic bond on demand. Complementary reactive groups are provided via well-defined short linear polymer chains of maleimide functionalized poly(lauryl methacrylates) **P** ($M_n = 6,000–8,600$ g mol$^{-1}$) synthesized by atom transfer radical polymerization with narrow dispersity ($Đ = 1.17–1.26$) and with a variable amount of maleimide that can be adjusted modularly to achieve the best crosslinking density (for synthesis see Supplementary Methods)[28]. The enhanced mobility necessary for a surface rearrangement and effective reflow of the material—in particular in thin film coatings—is accomplished by introduction of dodecyl side chains in the polymer backbone warranting a low glass transition temperature of $T_g = -44$ °C (Supplementary Fig. 1)[29,30] and additionally providing optical transparency above 350 nm (Supplementary Fig. 2).

**Controlling the healing ability in scratched thin films.** Local control over the thermal healing process was investigated using scratched thin films of **X@P$_{ON}$** (Fig. 2) prepared from **X$_{ON}$**

carrying four furyl groups and **P** (11 mol% maleimide content) mixed in a ratio to assure 0.7 equivalents of furyl groups per maleimide unit to minimize the optical density. The crosslinking process was determined to yield roughly 88% conversion of the furyl groups by ultraviolet/visible spectroscopy (Supplementary Fig. 3 and Supplementary Note 1) and qualitatively confirmed by solid state $^1$H-NMR measurements (Supplementary Fig. 4 and Supplementary Note 2) as well as small-angle X-ray scattering (SAXS) techniques (Supplementary Fig. 5 and Supplementary Note 3). A model scratch of the unaltered cross-linked material is healed by heating above 120 °C ($> T_g$) indicating the thermal healing capability of the dynamic polymer network (Fig. 2a,e). The thermoreversibility of this dynamic system is confirmed in Supplementary Figs 6–8. Contrarily, illumination of a part of the scratched thin film with ultraviolet-light locally suppresses the thermal healing by formation of a static polymer network (Fig. 2b,c, right side). Healing capability (Fig. 2d,e) is again re-established on illumination with visible light (Fig. 2c,d). This process can be repeated successfully over several cycles indicating complete reversibility of all involved steps. Detailed scratching tests also including blank sample **P** without crosslinker can be found in Supplementary Figs 9 and 10.

**Generation and characterization of a static polymer network.** The selective transition from the dynamic to the static state (Fig. 2a,b) was carried out by illumination of the scratched cross-linked network **X@P$_{ON}$** with ultraviolet-light (365 nm LED) through a photomask generating **X@P$_{OFF}$** locally. The locked network **X@P$_{OFF}$** clearly contains ring-closed crosslinker as indicated by the colouration of the exposed area. The formation of **X@P$_{OFF}$** is confirmed by solid state ultraviolet/visible spectroscopy where the characteristic absorption band of the ring-closed Diels–Alder adduct—as compared with the respective small molecule reference compound in solution (Supplementary Fig. 11, Supplementary Note 4)—can be retrieved in the illuminated area (Fig. 3a). Moreover, locking **X@P$_{ON}$** to **X@P$_{OFF}$** can be followed by Fourier transform infrared (FT-IR) spectroscopy, where a shift of the characteristic out of plane C–H bending mode of the vinylic group in **X@P$_{ON}$** at 868 to 854 cm$^{-1}$ corresponding to the allylic

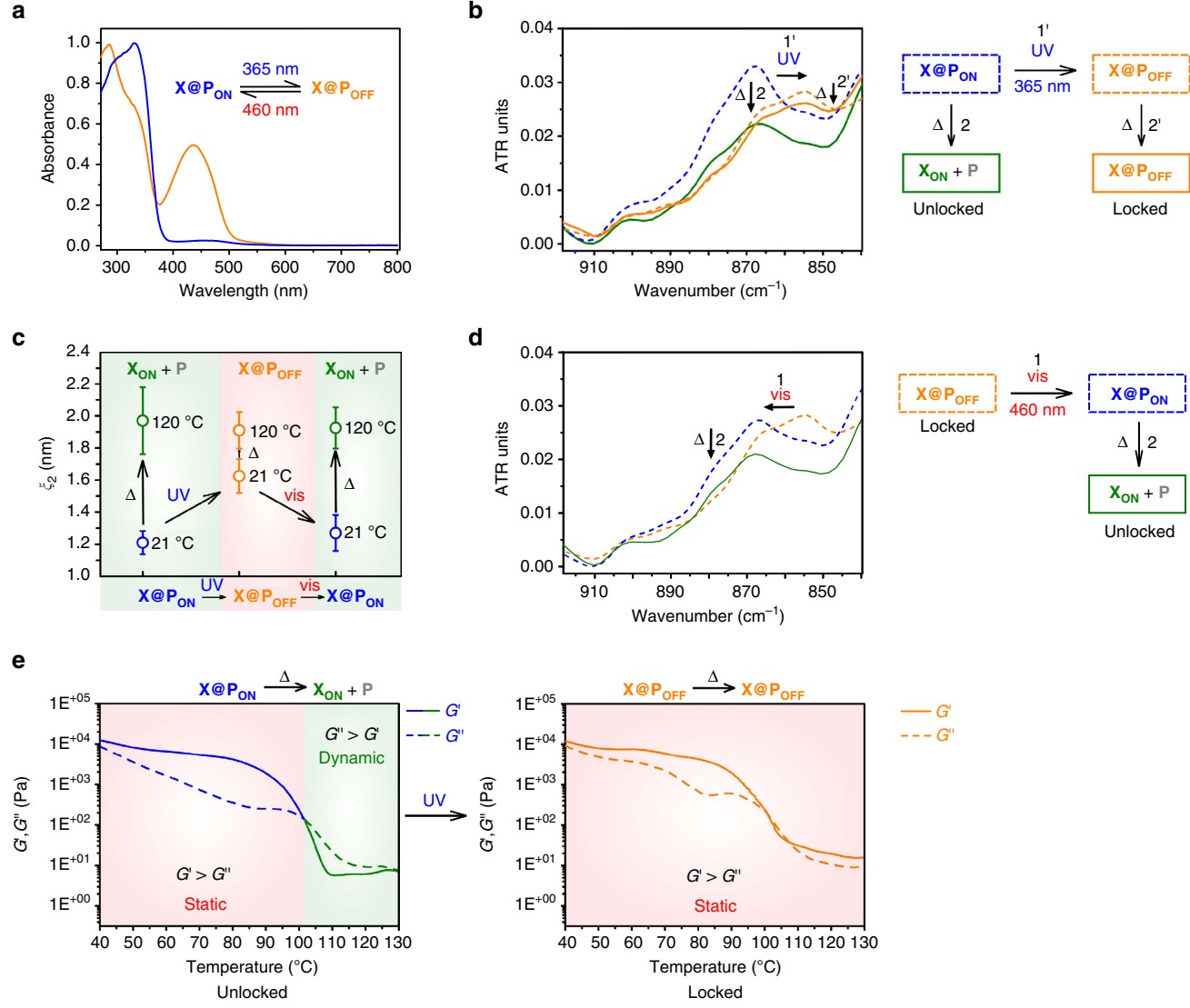

**Figure 3 | Characterization of individual states during a light-controlled healing cycle.** (**a**) Normalized solid state ultraviolet/visible spectra of **X@P$_{ON}$** on illumination with a 365 nm LED for 8 min to form the ring-closed isomer **X@P$_{OFF}$** (83% photoconversion in photostationary state, Supplementary Fig. 11 and Supplementary Note 4). (**b**) FT-IR spectra of **X@P$_{ON}$** and **X@P$_{OFF}$** recorded at 25 °C and after heating to 124 °C for 5 min in vacuum. (**c**) Mesh sizes of **X@P$_{ON}$**, **X$_{ON}$ + P** and **X@P$_{OFF}$** measured by SAXS (scattering curves: Supplementary Figs 5, 12, 13 and 20, overview over mesh sizes: Supplementary Tables 1–5, further explanation: Supplementary Note 3). (**d**) FT-IR spectra of **X@P$_{OFF}$** (recorded at 25 °C) and **X@P$_{ON}$** regenerated on illumination with 460 nm LED for 105 min (recorded at 25 °C and after heating to 124 °C for 5 min in vacuum). (**e**) Temperature dependence of shear storage modulus $G'$ and shear loss modulus $G''$ of **X@P$_{ON}$** and **X@P$_{OFF}$**.

group in **X@P$_{OFF}$** is clearly visible (Fig. 3b, blue dashed lines to yellow dashed lines)[31]. Interestingly, on ultraviolet-illumination the mean polymeric mesh size increases from $\xi_2 = (1.21 \pm 0.07)$ nm for **X@P$_{ON}$** to $\xi_2 = (1.61 \pm 0.10)$ nm for **X@P$_{OFF}$** as determined by SAXS measurements (Fig. 3c; Supplementary Figs 5, 12 and 13). At first, this finding appears to be counterintuitive but we attribute it to a more efficient photochemical ring-closure in the smaller meshes due to the higher optical density of the photoswitch. This leads to preferential 'disappearance' of the smaller meshes as they fall below the SAXS lower size detection limit of > 0.7–0.8 nm effectively widening the mesh size (Supplementary Table 1 and Supplementary Note 3).

**Conditional repair of the polymer network.** In addition to the revealing visual comparison of the unlocked, dynamic and the locked, static networks' response to heat with healing occurring

only in the non-illuminated area, (Fig. 2b,c) the structural molecular basis of these macroscopic changes can be monitored by FT-IR spectroscopy. While heating of **X@P$_{ON}$** above 120 °C results in a significant decrease in the vinyl group's C–H bending signal clearly indicating the de-crosslinking process, (Fig. 3b, blue dashed line to green solid line) heating **X@P$_{OFF}$** to the same temperature retains the spectral features demonstrating the successful inhibition of the retro Diels–Alder reaction by light (Fig. 3b, yellow dashed line to yellow solid line). Examining the thermo-rheological behaviour of **X@P$_{ON}$** reveals a typical drop of the shear storage modulus $G'$ below the loss modulus $G''$ above 100 °C that is well documented for Diels–Alder cross-linked polymers[32,33] and verifies the dynamic de-crosslinking process (Fig. 3e, blue to green lines). In stark contrast, **X@P$_{OFF}$** retains its static viscoelastic properties on heating, that is, $G' > G''$ regardless of the temperature, providing additional proof for the successfully inhibited retro Diels–Alder reaction on the ultraviolet-

illuminated side (Fig. 3e, yellow lines, Supplementary Fig. 14). Monitoring of a number of distinct vibrational modes in comparison to small reference molecules by FT-IR and further SAXS as well as differential scanning calorimetry (DSC) measurements strongly underline this result (Supplementary Figs 7, 12, 13 and 15–17, Supplementary Tables 2 and 3, and Supplementary Notes 3 and 5).

**Restoration of the polymer network's dynamic character.** Unlocking $X@P_{OFF}$ to $X@P_{ON}$, thus restoring the polymer's dynamic character and its healing ability, (Fig. 2c,d) is realized by illumination with visible light (460 nm LED) and can easily be verified by the naked eye through the apparent decolouration of the material. Due to the rather slow ring-opening, no special precautions have to be taken when handling the locked network $X@P_{OFF}$ under ambient, low intensity sunlight over a time frame of several hours (Supplementary Fig. 18 and Supplementary Note 6). The ring-opening process can be followed in more detail by monitoring the restoration of the vibrational band at $868\,cm^{-1}$ using FT-IR spectroscopy (Fig. 3d, yellow dashed line to blue dashed line) as well as by the characteristic changes in the ultraviolet/visible spectrum (Supplementary Fig. 19). The observed recovery of the polymeric mesh's initial correlation length of $X@P_{ON}$ as monitored by SAXS adds further evidence to this step (Fig. 3c, Supplementary Figs 13 and 20, Supplementary Table 4 and Supplementary Note 3).

**Re-established healing capability.** Besides the clearly visible vanishing of the scratch on heating of the re-activated dynamic material (Fig. 2d,e), the regenerated healing ability of $X@P_{ON}$ is once again accompanied by a strong decrease of the indicative signal at $868\,cm^{-1}$ in the FT-IR spectrum (Fig. 3d, blue dashed lines to green solid lines) and a full widening of the polymeric mesh from $\xi_2 = (1.27 \pm 0.11)$ nm to $\xi_2 = (1.93 \pm 0.12)$ nm (Fig. 3c, Supplementary Fig. 20, Supplementary Table 5 and Supplementary Note 3). The light-induced local healing ability strongly suggests a chemical de-crosslinking/re-crosslinking process rather than physical melting to be responsible for the observed behaviour. This is further corroborated by rheology experiments at varying temperatures that provide insight into the kinetics of the (retro) Diels–Alder reactions, illustrating the competing effect of thermally driven de-crosslinking and thermally activated re-crosslinking (Supplementary Figs 8 and 21).

**Reversibility of all steps.** The material undergoing the light-controlled locking and de-locking cycle fully corresponds to $X@P_{ON}$ as generated by direct conversion (Fig. 2a,e) in every analytical method we employed, that is, optical (ultraviolet/visible, FT-IR), structural (SAXS), mechanical (temperature-dependent shear rheology) and thermal (DSC) properties. This highlights the reversibility of all steps involving illumination with ultraviolet and visible light as well as thermal treatment in the locked and unlocked state.

## Discussion

Our photoswitchable dynamic covalent material consists of linear polymethacrylate chains substituted with maleimide units that are connected to a network by a multivalent light-responsive furan crosslinker. The incorporation of the latter allows us to transform specific areas reversibly and robustly from a thermally healable dynamic to a static 'locked' polymer network and vice versa simply by locally applying light of different wavelengths. Extensive rheology, DSC, ultraviolet/visible and SAXS measurements provide sound evidence that the observed conditional healing truly has its origin in the formation and scission of

covalent bonds *via* the (retro) Diels–Alder reaction, as expected on the basis of the pioneering work by Wudl[11] and others[12]. We believe that the introduction of this additional switching stimulus solves one of the major drawbacks of healable materials that commonly attain their healing ability at the expense of their thermal integrity. As opposed to the method described by Rowan, Weder, and coworkers[17], this photoswitchable locking and unlocking moreover avoids continuous irradiation and thus light-induced material degradation. Moreover and in contrast to solely photohealable polymers, the combination of light activation with subsequent thermal healing provides better healability, presumably due to an enhanced material mobility necessary for repair[20]. While presented here in the context of thin film applications and functional coatings, we reason that our reactant system can be adapted to numerous classes of more complex, responsive polymer systems. The associated external control over local dynamic crosslinking and hence healing ability should aid the design of programmable, repairable soft materials for a multitude of future applications. This could include macroscopic properties, such as in healable paints, as well as microscopic features, such as in latent resists that can be carried through various processing steps until being activated to offer control over nanofabrication.

## Methods

**General remarks.** For syntheses and not previously mentioned characterization of crosslinker $X_{ON}$, the maleimide-functionalized monomer **MIMA** and furyl-protected, inactive linear polymer **fpP** as well as non-protected, active linear polymer **P** see Supplementary Figs 22 and 23 and Supplementary Methods.

**General procedure for preparation of X@P$_{ON}$ and X@P$_{OFF}$.** Copolymer **P** (1.0 equiv. maleimide side chains) and crosslinker $X_{ON}$ (0.7 equiv. furan termini) were dissolved in a minimum amount of dry THF in a 2.5 ml vial and subsequently drop-casted onto a glass slide positioned in a Schlenk flask under an argon atmosphere followed by subsequent evacuation. The evacuated Schlenk flask was heated at 130 °C for 90 min to evaporate remaining solvent and to anneal the polymer mixture, followed by thermal crosslinking at r.t. for additional 16 h in the evacuated Schlenk flask. For rheology, drop-casting was performed directly onto steel plates. Note that polymer network preparation has been optimized for thin films and no free-standing films (which would allow a classical tensile test) have been obtained. After thermal crosslinking, half of the bulk material was irradiated with a Roithner 365 nm-LED XSL-365-5E at 20 mA and 4.2 V LED in a distance of 3 cm orthogonal to the sample and used as the locked polymer network sample $X@P_{OFF}$ for further analysis. Furthermore, the locked and heated polymer film was irradiated with a LED Engin 460 nm Blue LED Emitter LZ4-00B208 at 2–3 mA and 12 V for 105 min to further investigate the reestablishment of the healability.

**General procedure for scratching tests of X@P$_{ON}$ and X@P$_{OFF}$.** Scratches were done with a scalpel in a controlled manner in the mm-scale, followed by masking half of the scratch and cross-linked polymer film $X@P_{ON}$ (prepared as described above) with aluminium foil. Illumination of the unmasked area was performed in a Schlenk flask under an argon atmosphere for 30 min by adjusting the above-mentioned Roithner 365 nm in a distance of 3 cm orthogonal to the glass slide to form $X@P_{OFF}$. After removal of the aluminium foil, the glass slide was heated on a Deben Enhanced Coolstage from ambient temperatures to 124 °C in a specimen chamber vacuum ($\sim$30–50 Pa) of a SEM TM-1000 control unit remaining at that temperature for 5 min. For reestablishment of the healability, the polymer film was irradiated with the above-mentioned LED Engin 460 nm-Blue LED Emitter for 105 min, followed by another heating process (same conditions as for locking the polymer). Optical micrographs as well as FT-IR spectra were recorded after each step.

**Instrumentation.** Optical micrographs were either acquired on a Bruker A670 Hyperion FT-IR microscope in the reflection mode with a visible objective, 40-times magnified ($\times 10$ ocular, $\times 4$ vis-objective, for micrographs in main article) or with the optical detection unit of the atomic force microscopy (AFM) facility (for micrographs presented in the SI). The white balance of all images was automatically adjusted with a macro written for the Fiji software, provided by the light microscopy facility of the Cambridge Institute for Cancer Research, UK. FT-IR spectroscopy was carried out on a Bruker Vertex 70v equipped with a Specac Golden Gate single reflection diamond ATR sample holder. Scans (number of scans: 128) were collected with a resolution of $4\,cm^{-1}$ from 4,000 to $400\,cm^{-1}$.

Baseline correction was performed by using spline interpolation in OriginPro 9.1, OriginLab Corp., Northampton, USA. Rheological measurements of $X@P_{ON}$ and $X@P_{OFF}$ were performed on a Modular Advanced Rheometer System (HAAKE MARS II) of Thermo Fisher Scientific GmbH, Karlsruhe. An oscillatory shear mode with parallel plate geometry and 20 mm-diameter disposable aluminium plates was utilized for all experiments. A constant deformation of 0.1% shear strain at a frequency of 1 Hz was used. For Fig. 3e and Supplementary Figs 14a and 23 experiments were done with a heating ramp rate of 0.02 K s$^{-1}$ from 40 to 160 °C by collecting data points every 10 s. A fixed value of 0.12 mm of sample thickness was set. For Supplementary Figs 8 and 21 a constant normal force of 0.2 N was applied for gap control. Heating ramps were conducted at ca. 0.5 K s$^{-1}$ utilizing a Peltier element and collecting data points every 7–15 s with three repetitions per data point. Data analysis were performed with the software HAAKE RheoWin 4.3. Rheological data were smoothed using the Adjacent-Averaging-Method in OriginPro 9.1, OriginLab Corp., Northampton, USA, except of graphs in Supplementary Figs 8 and 21b. Note that compression or tensile measurements could not be performed due to the film's high viscoelasticity and creep even at low temperature. SAXS measurements of $X@P_{ON}$ and $X@P_{OFF}$ were performed in a solid sample holder with a Kratky-type instrument (SAXSess from Anton Paar, Austria) at temperatures of $(21 \pm 1)$ °C and $(120 \pm 2)$ °C. The SAXSess has a low sample-to-detector distance (0.309 m), which is appropriate for short measurement times of 10 min. The measured intensity was corrected by subtracting the intensity of the empty sample holder with a 30 µm-thick aluminium foil. The scattering vector is defined in terms of the scattering angle $\theta$ and the wavelength of the radiation ($\lambda = 0.154$ nm): thus $q = 4\pi n/\lambda \sin\theta$. Deconvolution (slit length desmearing) of the SAXS curves was performed with the SAXS-Quant software. Curve fitting was conducted with the software SASfit. For solid state ultraviolet/visible spectroscopy of $X@P_{ON}$ and $X@P_{OFF}$ 3–5 µl of a solution of free crosslinker $X_{ON}$ (0.7 equivalent furan per maleimide unit) and the respective polymer in degassed THF were spin-coated on $1 \times 1$ cm quartz glass plates (thickness of plates: 1 mm; polymer film thickness approx. 2.0 µm, determined by AFM measurements in tapping mode). Spin coating was performed at a rotation speed of 100–150 rps with a spin coating time set to 60 s with a KLM spin coater SCC-200 from SCHAEFER Technologies Corporation (Langen, Germany) at room temperature. Thermal crosslinking was carried out for 16 h at r.t. Irradiation was performed directly in a Varian Cary 50 ultraviolet/visible spectrophotometer equipped with a Peltier thermostated cell holder at $25 \pm 0.05$ °C by adjusting the LED in a distance of 1 cm orthogonal to the quartz glass plate in the sample holder. A Roithner 365 nm-LED XSL-365-5E for ring-closing at 20 mA and 4.2 V and a LED Engin 460 nm-Blue LED Emitter LZ4-00B208 for ring-opening at 2–3 mA and 12 V were employed, both driven by a GW Instek GPD-3303S linear DC power supply.

**Data availability**. All data that support the findings of this study are available from the corresponding author on reasonable request.

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

## Acknowledgements

We thank J. Hildebrandt and J. Schwarz for upscaling the synthesis of precursor DAEs, S. Winterhalder and M. Nguyen Trung for upscaling the synthesis of some copolymers, S. Winterhalder for help in the synthesis of the small molecule reference compound, B. Kobin and J. Leistner for help in instrumentation adjustment, K. Rademann for providing the set-up for scratch healing tests (SEM, cooling stage, and vacuum chamber), AFM and portable ultraviolet/visible spectrometer, G. Scholz for measuring and interpreting solid state NMR spectra, and E. Pietsch for providing assistance in some rheology measurements. Generous support by the European Research Council via ERC-2012-STG-308117 (Light4Function) is gratefully acknowledged. R.G. is supported by the German Research Foundation (DFG) through a research fellowship (GO 2634/1-1). J.K., M.D.H. and U.S.S. thank the German Research Foundation (DFG) for support within the SPP 1568.

## Author contributions

A.F. prepared the photoswitchable crosslinkers, and the resulting polymer networks, carried out ultraviolet/visible, FT-IR, GPC and DSC measurements and their data analyses, as well as performed healing studies. A.F. and J.K. synthesized and analysed some maleimide-containing poly(lauryl methacrylate) copolymers and did initial healing experiments and DSC measurements. A.F., K.B.-J. and U.N. did mechanical testing and A.F. did data interpretation with assistance of A.F.T., M.D.H. and M.B. A.F.T. performed SAXS measurements (with assistance of K.B.-J. and A.F.) and data analysis. R.W. did

AFM measurements to determine thickness in spectroscopically thin polymer films and measured sunlight spectra with a portable ultraviolet/visible spectrometer, helped in implementation of scratch healing tests in the used set-up and helped in FT-IR data interpretation. A.F., R.G. and S.H. conceived the idea, designed the study with the help of M.D.H. and U.S.S., and wrote the manuscript. All authors discussed the results and edited the manuscript.

## Additional information

**Competing financial interests:** The authors declare no competing financial interests.

**Publisher's note**: 

