## [Peer Review File · Nature Communications]

Reviewers' Comments:

Reviewer #2 (Remarks to the Author)

The authors have addressed the concerns raised by reviewers and the changes are appropriate and adequate. I recommend the acceptance of the paper in Nature Communications as is.

Reviewer #3 (Remarks to the Author)

Hecht et al. describe a polymeric network that can be selectively activated or deactivated for healing capabilities based on exposure to light and/or heat. The chemistry is based on a reversible Diels-Alder reaction that can be selectively triggered. They use a scratch test on the material to demonstrate healing ability. The authors state in their abstract that the material could have applications in photolithography, responsive coatings or microfabrication.

The subject matter is interesting but there are points that should be addressed before publication. My recommendation is that the following additions/modifications are made:

- 1) The thickness of the films was measured with AFM. Please include data on the thickness of the films at the scratch site before and after healing compared to other sites of the film to provide evidence of healing.
- 2) Please include a statement perhaps at the end of page 8 in the Discussion about how this is expected to have an impact in the different application areas discussed in the abstract. Healable paint? Carrying through fabrication steps as a feature until triggered to offer control during microfabrication?

Point-by-point reply to referee comments and implemented changes in NCOMMS-16-18166-A:

Reviewer #2:

The authors have addressed the concerns raised by reviewers and the changes are appropriate and adequate. I recommend the acceptance of the paper in Nature Communications as is.

Reply: We thank the reviewer for recommending the acceptance of our manuscript in Nature Communications and hence for acknowledging the quality of our work!

Reviewer #3:

Hecht *et al.* describe a polymeric network that can be selectively activated or deactivated for healing capabilities based on exposure to light and/or heat. The chemistry is based on a reversible Diels-Alder reaction that can be selectively triggered. They use a scratch test on the material to demonstrate healing ability. The authors state in their abstract that the material could have applications in photolithography, responsive coatings or microfabrication. The subject matter is interesting but there are points that should be addressed before publication. My recommendation is that the following additions/modifications are made:

1) The thickness of the films was measured with AFM. Please include data on the thickness of the films at the scratch site before and after healing compared to other sites of the film to provide evidence of healing.

Reply: Indeed, we were able to determine a thickness of a spectroscopically thin (and optically not too dense) polymer film by AFM measurements in the tapping mode to amount to $d \approx 2.0 \mu\text{m}$. These films were used to quantify the Diels-Alder crosslinking reaction in the solid state with help of UV/vis spectroscopic measurements (see Supplementary Note 1 on page S25 in the Supplementary Information). However, to determine the thickness of the thicker films used for scratch healing tests, AFM turned out to be inappropriate as it has a height limitation of approximately $10 \mu\text{m}$. In contrast, we are assuming our polymer films to have a thickness of at least $100 \mu\text{m}$ since for rheological measurements a plate-to-plate gap of the rheometer of $120 \mu\text{m}$ was set according to the sample's thickness. For healing tests, in general ca. 70% of the sample's depth were scratched. Although the topology of the films could in principle be measured using different techniques/instruments, such as a profilometer or a laser microscope, the macroscopic differences of the scratch depth before and after healing are rather obvious from the optical micrographs.

2) Please include a statement perhaps at the end of page 8 in the Discussion about how this is expected to have an impact in the different application areas discussed in the abstract. Healable paint? Carrying through fabrication steps as a feature until triggered to offer control during microfabrication?

Reply: To underline the impact of our work in different application areas, we followed the reviewer's suggestion to discuss about potential applications and hence inserted the following sentence in the end of the final Discussion section on page 8 of the main article: "This could include macroscopic properties, such as in healable paints, as well as microscopic features, such as in latent resists that can be carried through various processing steps until being activated to offer control over nanofabrication."